# Do the Lower Body Strength Assessment Tests in the Spanish Navy Really Measure What They Purport to Measure?

**DOI:** 10.3390/ijerph20010049

**Published:** 2022-12-21

**Authors:** Mª Helena Vila, Iris M. de Oliveira, Francisco J. Burgos-Martos, Angel Martín-Pinadero, Irimia Mollinedo-Cardalda, José M. Cancela-Carral

**Affiliations:** 1Department of Sports’ Special Didactics, Universidade de Vigo, Campus A Xunqueira, s/n, CP36005 Pontevedra, Spain; 2Department of Functional Biology and Health Sciences, Universidade de Vigo, Campus A Xunqueira, s/n, CP36005 Pontevedra, Spain; 3Department of Physical Education, Military Naval Academy in Marín, CP36913 Marín, Spain

**Keywords:** physical fitness, body composition, muscle mass, women

## Abstract

The main objective of this research was to analyse the efficacy of lower body strength assessment tests in the Armed Forces Physical Assessment System. Secondly, it was to determine what relationship exists between the physical evaluation system of the Spanish Armed forces and standardized evaluation protocols (Gold standard). A total of 905 students enrolled in the military/civil bachelor’s degree (813 male and 92 female) participated in this study. The influence of the sex of the participants was studied through the student’s t-test for independent data, and the degree of association between variables was defined by Pearson’s correlation coefficient. The results present moderate correlations (r = 0.67, r = 0.66; *p* < 0.001) between the vertical jump test used by the Army and the power or elastic force tests commonly used in practice and in research. The results obtained reflect a moderate relationship between the gold standard tests and the tests used by the Army, which suggests that the tests currently used to assess lower body strength should be adapted to more objective measurement tools which would allow a better comparison between samples from different armed forces.

## 1. Introduction

There are a variety of occupations within the military which require unique skills and have different physical demands. High levels of physical fitness (PF) and specific training programs are increasingly recognized as important factors for success in physically demanding occupations [1,2]. The different tasks require various strength, anaerobic power, and aerobic capacity along with the nutritional intake necessary to cover the different energy demands [3,4,5,6]. In addition to improvements in task performance, high levels of PF correlate with a reduced risk of musculoskeletal injuries [5,7,8,9]. Reducing the incidence of injuries is essential in guaranteeing tactical preparation and the availability of personnel to carry out their professional duties. For this reason, improvements in individualization and monitoring processes carried out during the training of military personnel are key.

In military personnel, PF is traditionally measured from simple field tests such as push-ups, sit-ups, and pull-ups. However, the validity of field tests is questionable due to a reportedly weak connection between individuals’ field test performances and their actual ability to perform military jobs that require strength (lifting and carrying tasks) [10]. Jumping ability has been shown to correlate with performance in military simulations [11]. Previous research has shown that Counter Movement Jump (CMJ) height is related to the performance of tactical work [12,13]. In the field, jumping tests are widely used for the assessment of lower body muscle strength [14], the most commonly used being the Squat Jump (SJ), CMJ and Avalakof test [11,15]. In the study presented by Vantakaris et al. [16] it is suggested that adding power-oriented strength work in military training could improve the ability to perform military tasks. It is therefore essential to study, develop and implement more adequate field tests, which in turn are feasible for use across large samples [17]. One of the objectives of the armed forces is to produce resilient military personnel, who are physically capable of performing the tasks entrusted to them. To our knowledge, no previous research has been carried out to analyse muscle power in soldiers of any of the Spanish armed forces (land, air, and sea). Therefore, the main objective of this research was to analyse the relevance of the lower body strength assessment tests used in the Army Physical Assessment System (SEFIET), and to determine the relationship between the SEFIET with standardized evaluation protocols. 

## 2. Materials and Methods

### 2.1. Study Design and Participants

This is quantitative research with a cross-sectional descriptive-correlational design. The sample is made up of students who were enrolled in the Naval School Marín (NSM) and attached to the University of Vigo during the academic years 2019/2020 and 2020/2021. A total of 905 students enrolled in the military/civil bachelor’s degree at the NSM (813 male and 92 female) participated in this study, aged between 18 and 37. The research complies with the Declaration of Helsinki regarding treatment and data collection and with the Spanish Organic Law on the Protection of Personal Data, and was approved by the Ethics Committee of the Faculty of Education and Sports Sciences of the Pontevedra, whose code is 03-719. Informed consent was obtained from all subjects involved in the study.

### 2.2. Assessment 

An anthropometric assessment of body mass (kg; Tanita TBF300^®^, Tanita Corp., Amsterdam, The Netherlands) and height (cm; Model HR001^®^, Tanita Corp., Amsterdam, The Netherlands) was made, as well as the Body Mass Index (BMI) which was calculated by dividing body mass by height squared (kg/m^2^). The anthropometric measurements were taken according to the protocols of the International Working Group of Kinanthropometry (ISAK) [18]. For the study of body composition, fat free mass (FFM), fat mass (FM), bone mass (BM), total body water (TBW), intracellular fluids (ICF), and skeletal muscle index (SMI) were calculated from bioelectrical impedance analysis (BIA) data, using the Tanita TBF300^®^ model (Tanita Corp., Amsterdam, The Netherlands).

The SEFIET has been established as an instrument for physical assessment which provides information for decision-making in physical education, used throughout the different Armed Forces and their training support systems. A Physical Condition Assessment (PCA) was carried out on Armed Forces personnel by means of tests performed following the technical instructions of the General Physical Condition Test [19], focused on the assessment of strength of the lower body, which were:50 m race. The race is started from a standing position and without hand support, from a defined line on the ground. At the “Go” signal, the stopwatch is started, and the athlete starts running and continues until the finish line is crossed, at which point the stopwatch is stopped. Time is recorded in seconds to the nearest tenth of a second.Vertical jump. A vertical jump is performed by flexion-extension of the legs, starting from a standing position. An initial baseline measurement is taken, for which the individual stands next to a wall with their feet shoulder width apart, and both arms are raised to a vertical position whilst keeping the position of the shoulders horizontal. The initial reading is the highest point marked at this position by the fingertips, which have been previously dipped into climbing chalk. Next, a vertical jump is made, and at the highest point, the participant tries to make a mark on the ruled surface of the wall with the tips of their fingers. They have two attempts. Height is recorded in centimetres.

For the complementary assessment of muscle power, two gold standard tests were used: the Squat Jump (SJ), and Countermovement Jumps (CMJ). The particiapants underwent a training period prior to performing the jumps. Data were recorded with a Chrono Jump Bosco System platform (Chronojump Bosco test, Barcelona, Spain). Two trials were performed for each test and the best score was recorded. A two-minute break was allowed between attempts, and between the SJ and CMJ. Both tests were performed with the participants keeping their hands on their hips, and with their legs fully extended during the airborne phase. Jumping technique was visually inspected by an expert evaluator. Height was recorded in centimetres.

### 2.3. Procedure 

The evaluations were carried out in three different facilities: on an athletics track (50 m), in a multipurpose room (vertical jump) in the NSM and in a sports hall at the Faculty of Education and Sports Sciences (University of Vigo). The assessment was carried out for one week, from 8:00 to 9:30 in the morning during the first three months of the years 2020 and 2021. An anthropometric assessment was carried out in the multipurpose room before the warm-up. Before carrying out the physical tests, the participants performed 15 min of a specific warm-up, tailored to the physical condition tests that were going to be carried out. The physical fitness assessments were undertaken by the armed forces´ own personnel from the Physical Education Department, who had been carrying out these assessments for at least two years. The anthropometric assessment was performed by ISAK technicians. The assessment of the SJ and CMJ was performed by researchers specialized in these assessments.

### 2.4. Statistical Analysis 

Descriptive analyses are presented as mean, median, standard deviations, minimum and maximum, stratified by sex. The normality of the sample was checked through the Kolmogorov–Smirnov test (*p* > 0.05). A comparative analysis was carried out between men and women through the student’s *t*-test for independent data, with the aim of analysing the behaviour of each of the variables analysed according to the sex of the participant. In order to compare the results of the tests undertaken with the SEFIET protocol and those with the gold standard of each test, a correlational analysis was carried out through the Pearson correlation coefficient (r). The following criteria were adopted to interpret the magnitude of the correlations: r < 0.1, trivial; 0.1 < r < 0.3, small; 0.3 < r < 0.5, moderate; 0.5 < r < 0.7, large; 0.7 < r < 0.9, very large; and r > 0.9, almost perfect [20]. The significance level was set at *p* < 0.05. Statistical analyses were performed using IBM-SPSS v.25 (IBM Co., Armonk, NY, USA).

## 3. Results

The mean values of the anthropometric characteristics and body composition of the military population are shown in Table 1. Women present a higher average age and in the rest of the anthropometric and body composition variables analysed, their values are smaller than those presented by the men, except for FM, these differences being significant in all cases. This is also maintained for the physical tests, where women present lower marks, and with significant differences to those obtained by men (Table 2).

The correlations presented by the two gold standard tests of strength, SJ with respect to CMJ are (r = 0.92), as can be seen in Figure 1, this behaviour being the same for both men and women.

The analysis of the degree of correlation between the 50 m run test with the levels of explosive strength and explosive elastic strength, reveal correlation coefficients of r = 0.71 and r = 0.68, respectively. Similarly, the analysis of the degree of correlation between the vertical jump field test with the levels of explosive strength and explosive elastic strength, reveal correlation coefficients of r = 0.67 and r = 0.66, respectively (see Figure 2). In both cases (50 m run and vertical jump), the degree of correlation was identical for both men and women. 

The correlational analysis between the field tests (50 m run and vertical jump) and MM levels were “r = 0.44, r = 0.46”, respectively, as shown in Figure 3.

## 4. Discussion

The objective of this study was to analyse the relevance of the lower body strength assessment tests used in the Armed Forces Physical Assessment System. In the present study, it was identified that the vertical jump test used by the Spanish armed forces to assess the explosive strength of the lower body does not present strong correlations with the explosive or elastic explosive strength tests which are commonly used in current practice and research [21]. Regarding the body composition variables, no strong correlations were recorded in the vertical jump variable. These results indicate a need to consider some changes to the way strength is assessed within the protocols designed by the Spanish armed forces. 

The current data support and reinforce the proposal that the current vertical jump tests have limitations with respect to the physical capacity they assess. This means that coaches and researchers need to rethink the lower body strength assessment tests, since the correlation values with the manifestation of explosive or elastic explosive force are only moderate. Therefore, it appears there is a need to progress to more adequate tests with a higher degree of reliability and validity [10,22]. Different devices or validation tests with new tools or protocol proposals can present correlations of greater than 0.9 [21]. There are currently mobile devices with high percentages of validity [21,23,24], in addition to different force platforms that would make the test change viable. Another positive aspect for the change would be the possibility of comparing the values of a Spanish armed forces sample with other populations of similar characteristics, which would help to improve or update the reference values for the evaluations carried out on this particular physical capacity. 

Coaches responsible for physical training in the military can use these data to be able to further fine-tune their assessments and control over lower body strength workload. Another aspect that backs up the test change is that the CMJ is a validated variable for monitoring performance adaptations, fatigue, and injury risk [15]. In the study presented by Merrigan et al. [15], they posit that the vertical jump restricts practitioners to a single metric (vertical jump height) but fails to provide more valuable information on movement strategies and relevant weaknesses, indicating that two subjects may have the same jump height but different neuromuscular strategies, which is why it is important to monitor strength-velocity metrics.

The results also confirm the existence of a gap between the values for strength and body composition of the men and women cadets analysed. These results are in line with the study by Aandstad et al. [17], conducted with Norwegian cadets. If the vertical jump values between both samples are compared, both the male and female cadets in this study jump higher than their Norwegian counterparts. Regarding body composition, however, Norwegian cadets are taller, heavier and have a slightly higher BMI, except for the women, who have similar values. Studies identify strength as an important physical capacity in fitness performance, and also that it correlates with a reduced risk of musculoskeletal injuries [5,7,8,9]. In the study proposed by Heebner et al. [25], they indicate that deficiency in knee strength is a risk factor for musculoskeletal injuries in a military population with highly demanding roles. 

The main limitations have been that the number of women participating in the study was small, which prevented us from carrying out a more in-depth analysis of their body composition and strength parameters. Additionally, including women introduces the phase of the menstrual cycle as an additional variable in this assessment. 

## 5. Conclusions

In conclusion, the results of this study provide practical information for researchers and professionals with a view to considering a change from the use of the vertical jump test to the SJ or CMJ as methods of assessing the strength development of military personnel. A better, more accurate assessment of the conditional aspects of the military personnel would allow better decisions to be made.

## Figures and Tables

**Figure 1 ijerph-20-00049-f001:**
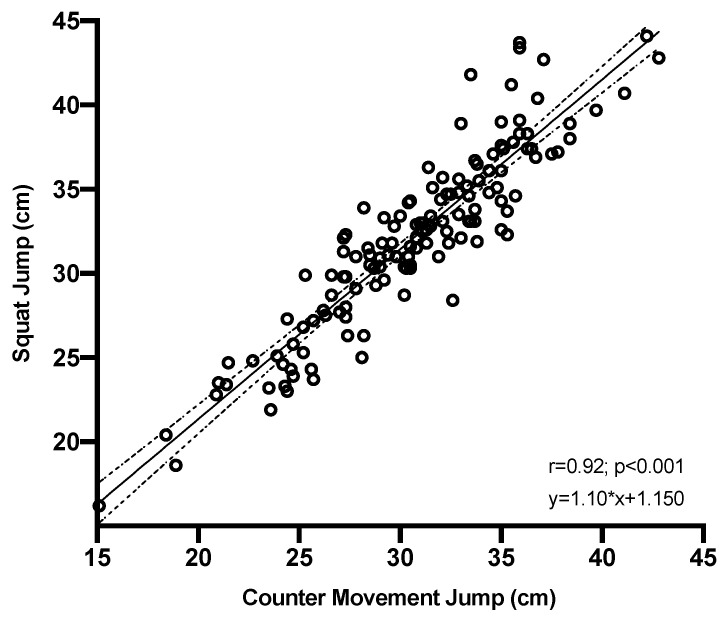
Correlation between elastic force and explosive elastic force.

**Figure 2 ijerph-20-00049-f002:**
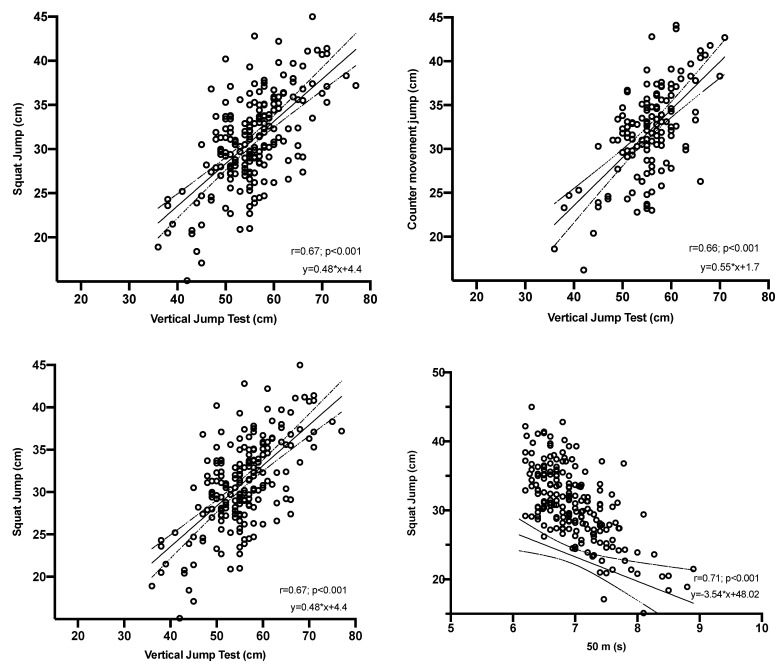
Correlations of the conditional variables with the vertical jump test.

**Figure 3 ijerph-20-00049-f003:**
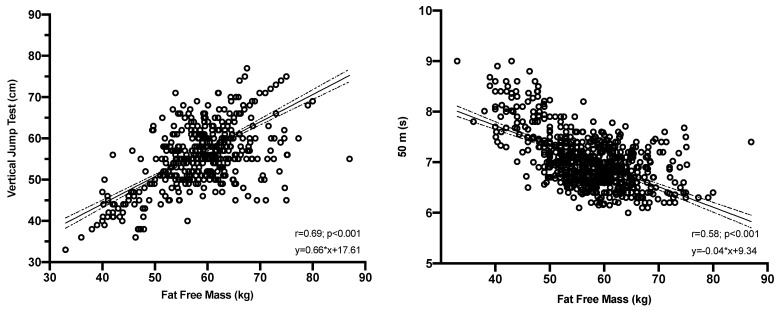
Correlation between Fat Free Mass and the “vertical jump” and “50 m run” field tests.

**Table 1 ijerph-20-00049-t001:** Participants´ anthropometric test stratified by sex.

	Men (n = 813)					Women (n = 92)					
	Mean	SD	Median	Min	Max	Mean	SD	Median	Min	Max	*p*
Age (years)	21.22	2.92	21.00	17.00	37.00	21.90	4.27	21.00	18.00	36.00	0.178
Body mass (kg)	73.89	8.68	72.40	56.00	104.60	60.65	5.86	60.10	50.10	73.20	<0.001
Height (cm)	177.14	67.3	177.00	161.00	198.00	165.37	4.93	165.00	157.00	178.00	<0.001
FFM (kg)	60.28	5.93	59.50	45.70	87.00	44.77	3.03	45.00	40.00	51.60	<0.001
FM (kg)	10.44	3.98	9.90	2.90	25.60	13.48	3.87	12.90	5.90	21.70	<0.001
BM (kg)	3.16	0.29	3.10	2.50	4.40	2.40	0.15	2.40	2.20	2.80	<0.001
TBW (kg)	46.51	4.21	46.10	36.00	66.20	34.08	2.34	34.20	30.40	39.10	<0.001
ICF (kg)	28.55	2.90	28.20	21.30	42.30	20.33	1.44	20.50	17.80	23.70	<0.001
BMI (kg/m^2^)	23.52	2.11	23.50	18.30	31.20	22.21	2.29	22.00	18.40	28.20	<0.001
SMI (cm^2^/m^2^)	31.86	4.30	31.38	22.68	49.38	21.07	2.59	21.37	16.46	26.41	<0.001

Legend: FFM: free fat mass; FM: fat mass; BM: bone mass; TBW: total body water; ICF: intracellu-lar fluids; BMI: body mass index; SMI: skeletal muscle index. SD: standard deviation; Min: Minimum; Max: Maximum.

**Table 2 ijerph-20-00049-t002:** Participants’ physical fitness test stratified by sex.

	Men (n = 813)			Women (n = 92)			
	Mean	SD	Median	Mean	SD	Median	*p*
Vertical Jump test (cm)	56.81	6.11	56.00	44.65	5.13	44.00	<0.001
50 m (s)	6.92	0.38	6.90	8.00	0.45	8.00	<0.001
SJ (cm)	31.97	4.75	31.90	23.10	3.45	23.60	<0.001
CMJ (cm)	32.94	4.68	32.85	23.53	3.76	24.30	<0.001

Legend: SJ: Squat Jump; CMJ: Countermovement Jumps; SD: standard deviation; Min: Minimum; Max: Maximum.

## Data Availability

Not available.

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
