# Peer review of "Do the Lower Body Strength Assessment Tests in the Spanish Navy Really Measure What They Purport to Measure?"

_ijerph, 2022, doi:10.3390/ijerph20010049_

Round 1

Reviewer 1 Report

This study is interesting regarding the efficacy of lower body strength assessment tests in the armed forces' physical assessment system.
There is a significant concern regarding the causality strength of this study; the authors should clarify that they can not go further than the correlation level. Otherwise, they should introduce a complete diagnostic test design.
On the other hand, the objective in real life should be the evaluation of physical performance free of injury, not only the correlation values with the gold standard. Therefore, the results of this study should be evaluated in further investigations that include the physical performance free of injury.

Author Response

Thank you for your review which has allowed us to improve the quality of the article. Attached I send you a file with the answers

Reviewer 2 Report

This study associates the results of physical tests from the Spanish Navy. The only innovation of this report is the sample. The major strength is the sample size. However, it seems that the authors got lost during the writing of the work. In the introduction, the authors bring up some interesting sentences “However, the validity of such muscular fitness field tests is questionable due to a reportedly weak connection between individuals´ field test performances and their actual ability to perform military jobs that require strength (lifting and carrying tasks)”. This is important and improves the innovation of the study. However, they left that aside and performed two tests commonly applied to other samples; this does not detract from the innovation of its sample, but it does not meet the expectation provided in the introduction.

Although the study cannot be redone, the authors should attempt to further explore their discussion and explain why these tests are potentially better than those already in use (PCA). Additionally, reservations can be raised regarding the tests chosen (SJ and CMJ) to associate with those applied in the PCA. In general, a lot of work needs to be done to amplify the importance of the study. Many studies have already been published with this same objective, but they have either been explored or mentioned.

Introduction

Line 51 – Change the term “explosive” here and throughout the manuscript. Simplify to muscle power.

Last paragraph – The key points of the introduction were lost in the last paragraph. For instance, it is not clear how the “relevance” will be tested. Moreover, what tests authors are referring to as the gold standard? Lastly, no hypotheses were presented.

Materials and methods

This section is confusing. The authors did not provide some “experimental design” section explaining the steps. Some well-elaborated figure is mandatory to improve clarity.

Further, the authors must clarify why they used SJ and CMJ to validate the PCA tests. One of these tests is the 50m race, which has a different motor pattern than the jump. The excuse of “strength of the lower body” is not enough.

 Line 68 – change “weight” to “body mass”.

Line 68 – 71 – Improve the clarity here. The “was” seems to refer to BMI, but what about body mass and height?

Line 96 – Both SJ and CMJ are largely used in the sports sciences. However, the proper scientific basis must be presented since you input the gold standard term.

Line 119 – It was verified but was it confirmed?

2.4 – Does not bring the results in this text (Tables and Figures)

Line 128 – The interpretation of correlation is better based on the p-values. Although your reference is dealing with “behavioral sciences”, there are several cases in which “small” correlations are relevant, since there is a statistical tendency.

Results

Use . rather then , for decimal cases.

There are no decimal cases for age. If authors disagree, explain what ,22 means.

Were the correlations performed with all samples? If so, explain why you compared sexes since the correlations were performed with all the samples.

Line 132-133 – This information is not some result. You have already informed the male and female participants.

Line 144 – Forget the term “almost perfect”. Does not add anything to the study or to the results. The same for other sentences in that you have used “large”. Focus the correlation analysis based on the p-values.

Both Tables and Figures can be improved. I do recommend the use of specialized software, like Graph Pad Prism or similar.

Discussion

The discussion is poor and does not properly discuss the main results. Several sentences were focused on the anthropometrics comparison, for instance. There is a vast literature discussing the physiological and biomechanical dynamics of these measurements. You should bring the discussion referring to your sample. This is the only innovation of this study.  

Author Response

(The authors gave the same response as above.)

Reviewer 3 Report

I thank the authors for an opportunity to review their paper. The authors present a cross-sectional analysis comparing standard military testing protocols to gold standard testing protocols.

The first finding suggests significant differences between male and female participants, as expected. 

The second findings is that there are moderate correlations between measures of physical performance in the Army assessments and gold standard assessments such as the CMJ and VJ. 

However, muscle mass as measured by BIA revealed no significant associations with these measures of performance.

The authors conclude that while the tests of physical function are moderately correlated with those of the gold standard tests, there is room to optimise the tests to better suit the needs of the Army in identifying key performance characteristics.

Please see my comments below.

1. I have attached an edited PDF with strikethrough and comments for suggested replacements for words. Often the wording is excessive and would benefit from being made more direct. Please remove the words which have strikethrough and substitute suggest words provided on notes at each location.

2. In several sections (yellow highlight) - their are sentences which refer to University of 'XXX' which I believe is a typo and needs to be corrected in various locations.

3. Please spell out ISAK acronym at first usage (Line 115). 

4. In the footnote of the table, please add a brief interpretation of each measure - i.e. is a higher/lower BMI better or worse, and so on. 

5. The authors present a correlation between muscle mass (in absolute kg).

a. The muscle mass presented is likely 'lean mass' or 'fat free mass' if derived from BIA - this is different from muscle mass, and includes bone and organ tissue. It appears unlikely that an individual would have 90kg of muscle mass - however more likely that they would have 90kg of fat free mass if a heavy individual. Please clarify and correct terminology as necessary. 

b. If available, please provide the equations that are used to calculate fat free mass or muscle mass (whichever is relevant). 

c. As the presented measure is an absolute measure (kg), it would be more appropriate to present a skeletal muscle index (SMI) or fat free mass index which, similar to BMI represents the amount of muscle mass or fat free mass per height-squared. This will likely provide a more sensitive representation of and relationships between these measures and physical function. Please provide this analysis and graph. 

Thank you. 

Author Response

(The authors gave the same response as above.)

Round 2

Reviewer 1 Report

The authors answered the reviewers' observations and modified the manuscript.

Author Response

Thanks for your contributions

Reviewer 2 Report

Some of my questions were answered, however, the study still does not have an adequate rationale, and its results add little to the scientific community.

According to the authors, the main innovation is attributed to the determination of muscle power in Spanish soldiers. If this is the case, no particulars have been provided to support this objective. In this scenario, some questions arise: a) what is the particularity of Spanish soldiers for other armed forces? The literature has several studies that determined these parameters in military personnel from other nations. b) Again, how SEFIET relevancy will be tested. In summary, although a few changes have been made to the introduction, the rationale of the article still does not present innovation.

Authors adjusted writing questions, but provided responses that, at the very least, challenge scientific prerogatives when designing experiments. First, its introduction provides some subsidies for the formulation of the hypothesis. The fact that this is a "descriptive" study does not eliminate the need for a hypothesis. When the experimental design was conceived, what did the authors expect as a result? Anything? Second, I fail to understand how a descriptive study does not require a description of its experimental design. Finally, I recommend reflection when considering the magnitude of correlations in large groups of data. In fact, the r value infers the magnitude, but its interpretation is associated with the sample size.

Author Response

Point 1: Some of my questions were answered, however, the study still does not have an adequate rationale, and its results add little to the scientific community.

According to the authors, the main innovation is attributed to the determination of muscle power in Spanish soldiers. If this is the case, no particulars have been provided to support this objective. In this scenario, some questions arise: a) what is the particularity of Spanish soldiers for other armed forces? The literature has several studies that determined these parameters in military personnel from other nations. b) Again, how SEFIET relevancy will be tested. In summary, although a few changes have been made to the introduction, the rationale of the article still does not present innovation.

Response 1: We regret not being able to clearly explain the interest of the work. We are aware of the importance of force levels for the development of the professional work of the military, but our objective is not that. Our objective is to demonstrate that the methods/tests used to assess the levels of force in the Spanish army are not the most adequate. With this work we intend to give arguments to plant its change.

Authors adjusted writing questions, but provided responses that, at the very least, challenge scientific prerogatives when designing experiments. First, its introduction provides some subsidies for the formulation of the hypothesis. The fact that this is a "descriptive" study does not eliminate the need for a hypothesis. When the experimental design was conceived, what did the authors expect as a result? Anything? Second, I fail to understand how a descriptive study does not require a description of its experimental design. Finally, I recommend reflection when considering the magnitude of correlations in large groups of data. In fact, the r value infers the magnitude, but its interpretation is associated with the sample size.

Response 2: Dear reviewer, this is a descriptive study, and therefore there is no independent variable that the researcher can manipulate to have an effect on the dependent variable. This fact conditions the formulation of a research hypothesis. However, the following hypothesis could be formulated, which we could not verify: "The current tests used to evaluate the manifestations of force in the Spanish army are not adequate." Regarding the magnitude of the correlations, we must indicate that we are aware of the indicated ones, but the sample size is determined by the access places, so it is a defined parameter and not a variable.

Reviewer 3 Report

Thank you to the authors for making modifications to the manuscript. Most, but not all changes were implemented correctly. Please see my comments below. 

1. Skeletal muscle Index is not cm2/m2 - it is represented in kg/m2 - please correct this and any values within the table 1. Additionally, please make sure all table acronyms are correct - some appear to have strikethrough and or not have a correct acronym in the footnote. 

2. I maintain that the measure presented is not 'muscle mass' but rather 'lean body mass' which is inclusive of all tissue minus fat mass. It is biologically implausible that individuals with a normal BMI can have an average 'muscle mass' exceeding 60kg. Please refer to the two articles below for your reference.

https://journals.physiology.org/doi/full/10.1152/jappl.2000.89.1.81

https://onlinelibrary.wiley.com/doi/epdf/10.1002/ajhb.23102

Even in very large athletes - the SM and SMI is not as high as reported in your sample. The lean body mass correlates better with the values you report. Please adjust accordingly. 

3. Please refer to the manual for the machine you have reported to use (attached) - the values provided are Fat-free mass (FMM) which is not equivalent to muscle mass. Please provide a reference for that validation paper for the model you have used in your article.

4. Please re-do the graphs for SMI based upon the adjustments to calculations above - i.e. kg2 of SMM/m2 

Thank you.

Author Response

Point 3. Thank you to the authors for making modifications to the manuscript. Most, but not all changes were implemented correctly. Please see my comments below. 1. Skeletal muscle Index is not cm2/m2 - it is represented in kg/m2 - please correct this and any values within the table 1. Additionally, please make sure all table acronyms are correct - some appear to have strikethrough and or not have a correct acronym in the footnote.

Response 3: After reviewing several investigations and we understand that the units presented are correct. We indicate the bibliographical references consulted:

Kim EY, Jun KH, Kim SY, Chin HM. Body mass index and skeletal muscle index are useful prognostic factors for overall survival after gastrectomy for gastric cancer: Retrospective cohort study. Medicine (Baltimore). 2020 Nov 20;99(47):e23363. doi: 10.1097/MD.0000000000023363. PMID: 33217879; PMCID: PMC7676598.

Chang SW, Tsai YH, Hsu CM, Huang EI, Chang GH, Tsai MS, Tsai YT. Masticatory muscle index for indicating skeletal muscle mass in patients with head and neck cancer. PLoS One. 2021 May 10;16(5):e0251455. doi: 10.1371/journal.pone.0251455. PMID: 33970954; PMCID: PMC8109770.

We have corrected the acronyms.

Point 4. I maintain that the measure presented is not 'muscle mass' but rather 'lean body mass' which is inclusive of all tissue minus fat mass. It is biologically implausible that individuals with a normal BMI can have an average 'muscle mass' exceeding 60kg. Please refer to the two articles below for your reference.

https://journals.physiology.org/doi/full/10.1152/jappl.2000.89.1.81

https://onlinelibrary.wiley.com/doi/epdf/10.1002/ajhb.23102

Response 4: Thank you for your appreciation. After reviewing Tanita's documentation, and although in her description it says 'muscle mass', we completely agree with the review that this parameter refers to fat-free mass (FMM). For this reason, the parameter in the tables and graphs has been changed. Thank you very much for your support.

Point 5. Please refer to the manual for the machine you have reported to use (attached) - the values provided are Fat-free mass (FMM) which is not equivalent to muscle mass. Please provide a reference for that validation paper for the model you have used in your article.

Response 5: Thanks for the appreciation, the authors fully agree with the reviewer and we have proceeded to change the name of the variable.

Point 6. Please re-do the graphs for SMI based upon the adjustments to calculations above - i.e. kg2 of SMM/m2.

Response 6: With the references indicated regarding this variable and which have been clarified in response 3, we believe it is correct to keep the current graphs.

General comments

First, thank you for letting me read your interesting paper about the efficacy of lower body strength assessment tests in the Armed Forces Physical Assessment System. In general terms, the paper seems well written. The study determines the efficacy of lower body strength assessment tests in the Armed Forces Physical Assessment System and  to determine what relationship exists between the physical evaluation system of the Spanish Armed forces and standardized evaluation protocols (Gold standard). This topic is interesting, since it is necessary to analyze it. The methodological procedure is correct and shows great strength in the sample analyzed. In addition to being a specific sample, it is a large sample size. However, I recommend to carry out further statistical research to compare men and women according to the different variables and even to make groups by age range or experience (if this was taken into account).

Specific comments
Introduction

Line 37- “For this reason, improvements in the individualization and monitoring” I think that this sentences is incomplete.

Response: Thanks for the appreciation, we have proceeded to complete the sentence.

Line 45- Delete the coma “Previous research has shown that CMJ height is related to the 44 performance of tactical work [12,13], In the field,”

Response: Done

Line 47- change “avalakof” to “Abalakov test”

Response: Done

Line 58- Add a connector between sentences, e.g. therefore “Therefore, the main…”.

Response: Done

Line 60- connect the two objectives “Therefore, the objectives of this research were to analyse the relevance of the lower body strength assessment tests used in the Army Physical Assessment System (SEFIET) and to determine the relationship between the SEFIET with standardized evaluation protocols”.
Response: Done

Methods
Line 75- Delete “kg”.

Response: We have not deleted the unit of measurement, since the variables listed in that paragraph indicate the units of the other variables such as height.
Was the order of CMJ and SJ jumps randomised?

Response: We have not randomized the jumps. We have affected the recovery time between repetitions.

Results
Table 1- Correct the height values from Table.

Response: Done

Table 1 and 2- In the legend of the tables add all the abbreviations that appear in the table and if possible, in order. For example, sd (standard deviation) or min and max (standard deviation) are not added.

Response: Each abbreviation has been explained and placed in order of appearance in the tables.

Line 152- The correlation data given in the text is r = 0.91 and figure 1 shows r = 0.92.

Response: corrected.

Figure 1 is included in Figure 2. Leave only in one of the two figures. 

Response: corrected.

Do the figures show the correlation data for men and women? 

Response: Yes, the authors pooled the data, as there were no significant differences between men and women.

In figures 2 and 3 put the equation in a place where it can be seen better.

Response: The authors have changed the location of the equation

Figure 2

Figure 3

Discussion

Rewrite the objectives on the basis of previous corrections.

Improve the overall discussion. It only focuses on the fact that there is no high correlation between the vertical jump test and the CMJ or SJ. However, the first objective is not discussed. Furthermore, the differences obtained in men and women are not compared. I consider it necessary to add a paragraph on the limitations of the study itself.  

Response: We consider that the objectives have already been reformulated and are answered in a manner throughout the discussion. Regarding the comparison of men and women, where statistical differences have been identified, they have been pointed out. In the variables where no differences were identified, we proceeded to analyze the data in a general way. It was also not an objective of this study.

Study limitations have been included: As the main limitations, we can consider that the number of women participating in the study should be greater, in order to analyze a more in-depth analysis in terms of body composition and strength parameters. Also include the phase of the menstrual cycle as one more variable in this assessment.
